# FLEXTSF: A UNIVERSAL FORECASTING MODEL FOR TIME SERIES WITH VARIABLE REGULARITIES

## ABSTRACT

Developing a foundation model for time series forecasting across diverse domains has attracted significant attention in recent years. Existing works typically assume regularly sampled, well-structured data, limiting their applicability to more generalized scenarios where time series often contain missing values, unequal sequence lengths, and irregular time intervals between measurements. To cover diverse domains and handle variable regularities, we propose FlexTSF, a universal time series forecasting model that possesses better generalization and natively support both regular and irregular time series. FlexTSF produces forecasts in an autoregressive manner and incorporates three novel designs: VT-Norm, a normalization strategy to ablate data domain barriers, IVP Patcher, a patching module to learn representations from flexibly structured time series, and LED attention, an attention mechanism seamlessly integrating these two and propagate forecasts with awareness of domain and time information, enabling effective time series forecasting across varying regularities. Experiments on 12 datasets show that FlexTSF outperforms state-of-the-art forecasting models respectively designed for regular and irregular time series. Furthermore, after self-supervised pre-training, FlexTSF shows exceptional performance in both zero-shot and few-show settings for time series forecasting.

## 1 INTRODUCTION

Time series forecasting, the task of predicting future values based on historical observations, plays an indispensable role across numerous domains, including finance, manufacturing, retail, healthcare, and meteorology (Lim & Zohren, 2021; De Gooijer & Hyndman, 2006). Recently, advancements in large language models, which exhibit remarkable generalization ability to a range of language tasks(Zhao et al., 2023), have inspired a new trend of research focused on developing foundation models for time series forecasting. These models aim to deliver robust and effective performance across diverse time series datasets (Rasul et al., 2023; Ansari et al., 2024).

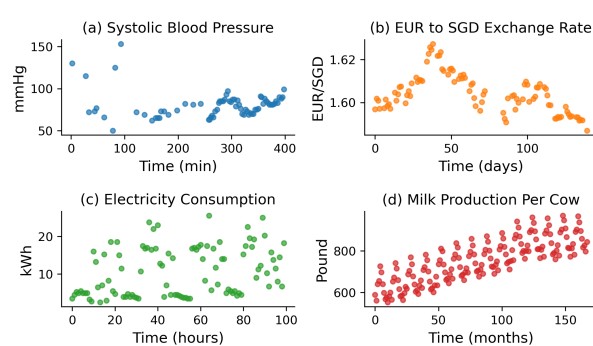

Figure 1: Illustration of time series from different domains with domain diversity and structural diversity.

Developing a foundation model for time series forecasting presents significant challenges due to the diverse characteristics of time series data (Figure 1). First, time series data contain a wide range of measurement types, such as systolic blood pressure,

exchange rate, and electricity consumption, each demonstrating different scales (e.g., 1-10, 50-150) and temporal granularities (e.g., minutes, hours, days, months). Such *domain diversity* leads to various temporal patterns that are difficult to be captured within a single model. Second, time series exhibit *structural diversity*, with missing values, varying sequence lengths, and irregular sampling time intervals. For instance, in Figure 1, the blood pressure observations depicted in (a) are sparse at the beginning but become denser over time due to the patient's deteriorating condition. In (b), some data is missing due to factors such as holidays. The time series in (d) shows a clear pattern, while the pattern in (c) is less obvious.

In this paper, we propose FlexTSF [1], a highly flexible time series forecasting model that not only can be broadly applicable but also performs well across time series with variable regularities in terms of various measurement types, completeness, lengths, or time intervals. The model employs a decoder-only architecture, where point-level time series input data is organized into patches, with each newly generated patch serving as the prediction for a bunch of future values (Woo et al., 2024; Das et al., 2024). Built on this backbone, the model incorporates several novel designs to address the diversity challenges of time series data.

To address the challenge of domain diversity, we propose *VT-Norm*, a value and time normalization strategy that unifies data with different characteristics by decoupling static domain information and dynamic temporal patterns. This strategy allows time series, regardless of scale or granularity, to be standardized, enabling the model to focus on learning temporal patterns and dependencies in a unified manner. Additionally, domain-specific features are extracted and structured into a "Leader node", which serves as a holistic reference and provides guidance for subsequent processing stages. This strategy enables the model's self-adaptation to different domains.

To address the challenge of structural diversity, we propose to enhance the patching module, which is originally designed to organize data points within a fixed window as a representation unit for model input (Nie et al., 2023). This approach implicitly assumes a uniform receptive field along the temporal dimension for each patch, a condition that only holds in regular and well-structured time series data. In light of this, we propose *IVP Patcher*, a continuous-time patching module capable of learning representations for time series patches that exhibit various structural diversity, such as missing values, arbitrary lengths, and various time intervals. The *IVP Patcher* is based on neural Initial Value Problem (IVP) solvers (Xiao et al., 2024), which model the temporal evolution process across arbitrary time intervals rather than the value patterns in fixed temporal window, providing effective support for both regular and irregular time series data.

We implement the forecasting process by proposing *LED Attention*, a causal self-attention mechanism that incorporates a **L**eader node, layerwise time **E**mbeddings, and a **D**ummy patch to iteratively process the time series for autoregressive prediction. To evaluate our designs, we deploy FlexTSF across 12 datasets according to train-validation-test split scheme to demonstrate its capability in tackling different time series characteristics from diverse domains and regularities. Furthermore, we pre-train FlexTSF on a large collection of datasets in an self-supervised manner, and evaluate its performance in zero-shot and few-shot settings, demonstrating its capability and potential to serve as a foundation model for time series forecasting.

We summarize our main contributions as follows:

- We introduce FlexTSF, a universal forecasting model for time series, offering a flexible solution to tackle time series with domain diversity and structural diversity.

- We propose novel designs: VT-Norm for domain self-adaptation, IVP Patcher for handling time series with variable regularities, and LED attention, which seamlessly integrates both to autoregressively propagate forecasts with domain and time awareness.

- We conduct extensive experiments showing FlexTSF's excellence in classic, zero-shot, and few-shot scenarios. Ablation studies confirm the positive impact of its three key components.

---

[1]Source code is at `https://anonymous.4open.science/r/FlexTSF-C3AE`.

## 2 RELATED WORK

There is an emerging trend of research training forecasting foundation models from scratch using time series data. In Table 1, we provide a technical comparison of five open-source foundation models with our FlexTSF. All these models are trained on large collections of time series data and have the capability for zero-shot prediction on new datasets. Specifically, ForecastPFN (Dooley et al., 2024) and DAM (Darlow et al., 2024) use pointwise time-value pairs as the input tokens within encoder-only models, but differ in their forecasting approaches: the former relies on time queries while the latter utilizes basis function composition. Lag-Llama (Rasul et al., 2023) and TimesFM (Das et al., 2024) employ decoder-only models, adopting autoregressive prediction, which is similar to next token prediction in LLMs, for training on time series data. MOIRAI (Woo et al., 2024) organizes time series into patches for multiple variables, leveraging an encoder-only model with masked patch prediction as the training objective.

| Name | ForecastPFN | Lag-Llama | DAM | MOIRAI | TimesFM | FlexTSF |
|---|---|---|---|---|---|---|
| **Tokenization** | Point-wise TV | Lag Feature | Point-wise TV | Patch | Patch | IVP Patcher |
| **Architecture** | Encoder-Only | Decoder-Only | Encoder-Only | Encoder-Only | Decoder-Only | Decoder-Only |
| **Forecast** | Time Query | Autoregression | Basis Functions | Masked Prediction | Autoregression | Masked Autoregression |
| **Variate** | Univariate | Univariate | Univariate | Multi/Univariate | Univariate | Univariate |
| **Probabilistic** | ✗ | ✓ | ✗ | ✓ | ✗ | ✓ |
| **Missing Values** | ✗ | ✗ | ✓ | ✗ | ✗ | ✓ |
| **Various Length** | ✓ | ✓ | ✓ | ✗ | ✓ | ✓ |
| **Irregular Interval** | ✓ | ✗ | ✓ | ✗ | ✗ | ✓ |

Table 1: Comparison between pre-trained time series forecasting models. "TV" indicates time-value pairs. ✓(or ✗) indicates the model (does not) naturally support this kind of data issue.

Existing studies on time series forecasting foundation models typically achieve the generalization capability by pre-training models on multiple time series datasets (Das et al., 2024; Rasul et al., 2023). However, these solutions for domain diversity are still immature. Some researchers alleviate this problem indirectly by using data from as many domains as possible for pre-training (Liang et al., 2024; Woo et al., 2024). The more types of data a model sees during pre-training, the fewer unfamiliar data types it will encounter in real applications. However, this paradigm is inefficient, and the time series data currently available for pre-training is not as extensive as the vast text corpora used to train large language models (Edwards et al., 2024; Touvron et al., 2023). Moreover, current research largely ignored the problem of structural diversity. Most of these models are designed for well-structured and regularly sampled time series (Ye et al., 2024), leaving them ill-equipped to handle the heterogeneous data structure issues often encountered in real-world scenarios. In contrast, as shown in Table 1, FlexTSF stands out by effectively handling various time series characteristics, such as missing values, varying lengths, and irregular intervals, making it more adaptable to diverse applications.

Additional related works on Transformers for time series forecasting, adapting LLMs for time series forecasting, and irregular time series modeling are provided in the Appendix A.3.

## 3 METHODOLOGY

### 3.1 PROBLEM FORMULATION

We consider a time series $S$ as a sequence of $M$ elements $S = \{(\boldsymbol{x}_i, t_i)\}_{i=1}^{M}$, where each element $(\boldsymbol{x}_i, t_i)$ consists of an observation $\boldsymbol{x}_i \in \mathbb{R}^v$ with $v$ variables collected at a specific timestamp $t_i$. This formulation is flexible to accommodate a diverse range of time series data characterized by various features, such as multiple variables ($v > 1$), different sequence lengths (varying $M$ across samples), irregular time intervals ($t_{i+1} - t_i \neq t_i - t_{i-1}$), and incomplete observations (some variables in $\boldsymbol{x}_i$ are missing), etc. Our target is to develop a foundation model for time series forecasting that can take any time series $S$ with different data characteristics as input and predicting its future values $\{\boldsymbol{x}_i\}_{i=M+1}^{M+H}$ over a subsequent time window $\{t_i\}_{i=M+1}^{M+H}$, where $H$ denotes the forecast horizon. A notation summary is available in the Appendix A.1.

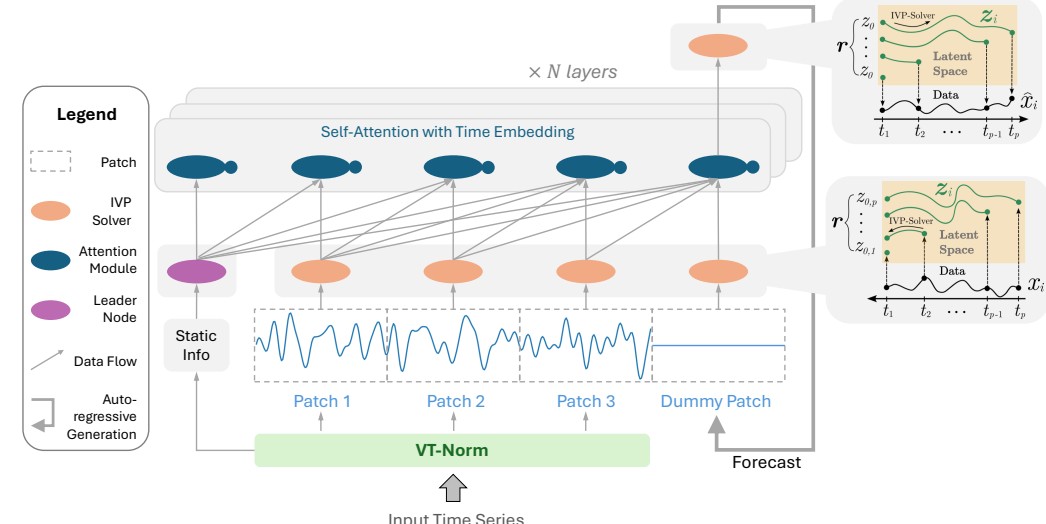

Figure 2: An overview of FlexTSF, which consists of the VT-Norm module, the IVP Patcher module, and a stack of layers with LED Attention.

## 3.2 PROPOSED MODEL: FLEXTSF

A dataflow overview of FlexTSF is illustrated in Figure 2. The process begins with VT-Norm and then segmenting the time series into patches and appending a dummy patch at the end. For each patch, an IVP solver evolves the latent states derived from all data points in the patch window backward in time to obtain a patch representation. These representations are subsequently processed by a stack of LED Attentions, with a Leader node encompassing statistical domain features extracted during normalization. After a forward pass, the representation of the final node, which corresponds to the dummy patch in the input, is passed to another IVP solver that evolves forward in time to generate the future time series values at specified time points. These predicted values replace the previous dummy patch, allowing the the autoregressive process to continue until all values within the forecasting horizon are produced. Next, we describe the details of each component.

### 3.2.1 VT-NORM

To address the challenges faced by foundation models due to diverse time series data characteristics, such as varying measurement types, scales, and sampling intervals, we propose VT-Norm to standardize both values and timestamps. Given the time series observation $\mathcal{X} = \{x_i\}_{i=1}^{M}$ from dataset $D$, we apply a two-step normalization scheme: global normalization and instance normalization. First, we calculate the mean and standard deviation of all variables in $D$ and use them to normalize $\mathcal{X}$ by subtracting the mean and dividing by the standard deviation. We refer to this process as global normalization and relevant statistics as global mean $\mu_g$ and global standard deviation $\sigma_g$. Next, before passing each time series into the model, we perform instance normalization (Nie et al., 2023) on the sequence itself, obtaining the instance mean $\mu_i$ and instance standard deviation $\sigma_i$. Let $G_g$ and $G_i$ denote the normalization operations using global and instance statistics, respectively. The whole normalization process for $\mathcal{X}$ can be described as: $\mathcal{X}' = G_i(G_g(X))$, where $\mathcal{X}'$ represents the normalized time series values.

For the corresponding timestamps $T = \{t_i\}_{i=1}^{M}$ of $\mathcal{X}$ in dataset $D$, we define the global time $\omega_g$ as the reciprocal of $D$'s frequency. For each sequence, we calculate the intervals between successive timestamps and obtain $\{\Delta t_i = t_{i+1} - t_i\}_{i=1}^{M-1}$. We then take the minimum of $\Delta t_i$ as the instance time unit $\omega_i$, and scale

the time intervals as $\Delta t_i' = \frac{\Delta t_i}{\omega_i}$. The new timestamps $T' = \{t_i'\}_{i=1}^M$ are calculated by summing the scaled time intervals: $t_i' = t_1 + \sum_{j=1}^{i-1} \Delta t_j'$.

This normalization mechanism effectively decouples static domain information from dynamic patterns. To summarize, for a time series sample $S$, we derive normalized values $\mathcal{X}'$ and normalized time indicators $T'$, along with six statistical features. We concatenate these features as: $L = [\mu_g, \sigma_g, \mu_i, \sigma_i, \omega_g, \omega_i]$. The vector $L$ is provided to the Leader node, while $\mathcal{X}'$ and $T'$ are fed into the IVP Patcher for subsequent processing.

### 3.2.2 IVP PATCHER

Unlike previous methods that are only applicable to regular time series and rely on splitting data into fixed temporal windows as patches (Nie et al., 2023), we propose IVP Patcher, which models the continuous evolution process of time series by solving initial value problems (IVP) (Xiao et al., 2024). This approach allows for us to derive temporal representations with arbitrary time intervals and allows for the handling of segments with varying lengths. A key idea behind our method is that time series values $x_i$ within a patch are discrete, indirect observations of an unknown continuous process (Xiao et al., 2024). To parameterize these processes with available observations, we use an ordinary differential equation of the form $\frac{dF(t)}{dt} = f(t, z_i)$, where $z_i$ is the hidden state of $x_i$. Given an initial condition $(t_0, z_0)$, the hidden state $z_i$ at $t_i$ can be computed using numerical methods (Chen et al., 2018; Biloš et al., 2021).

In the IVP Patcher, the time series input $\{(\boldsymbol{x}_i, t_i)\}_{i=1}^M$ is divided into non-overlapping patches. Each patch is represented as $V = \{(\boldsymbol{x}_i, t_i)\}_{i=p_s}^{p_s+p-1}$, where $p$ is the patch length (i.e., the number of observed data points within the patch), and $p_s$ is the start index of the patch. For simplicity, we take $p_s = 1$ as an illustration example, resulting in a patch $V = \{(\boldsymbol{x}_i, t_i)\}_{i=1}^p$. The output patch after LED Attentions shares the same patch length as the input patch, with $p_s$ and $p_e = p_s + p - 1$ denoting that start and end indices. Algorithm 1 describes the process of generating patch representations for input patches and producing predictions for the dummy patches.

For the input patches (**Part 1**), a linear layer maps each data point $\boldsymbol{x}_i$ to a latent state $\boldsymbol{z}_i$, which serves as the initial condition $(\boldsymbol{z}_i, t_i)$ for the neural IVP solver. The neural IVP solver starts at $t_i$ and evolves the state towards timestamp $t_1$, where observations begin:

$$\boldsymbol{z}_{1,i} = \text{IVPSolve}(\boldsymbol{z}_i, \Delta t_i), \qquad (1)$$

where $\Delta t_i = t_1 - t_i$.

---

**Algorithm 1** IVP Patcher

**Part 1**: Generate Patch Representations through Solving IVPs Backward in Time

**Input**: One Patch $\{(\boldsymbol{x}_i, t_i)\}_{i=1}^p$

**Output**: Patch Representation $\boldsymbol{r}$

1: $\{\boldsymbol{z}_i\}_{i=1}^p = \text{Linear}(\{\boldsymbol{x}_i\}_{i=1}^p)$

2: $\{\Delta t_i\}_{i=1}^p = t_1 - \{t_i\}_{i=1}^p$

3: $\{\boldsymbol{z}_{0,i}\}_{i=1}^p = \{\text{IVPSolve}(\boldsymbol{z}_i, \Delta t_i)\}_{i=1}^p$

4: $q(\boldsymbol{z}_0|V) = \text{Inference}(\{\boldsymbol{z}_{0,i}\}_{i=1}^p)$

5: $\boldsymbol{r} = \boldsymbol{z}_0 \sim q(\boldsymbol{z}_0|V)$

6: **return** $\boldsymbol{r}$

---

**Part 2**: Make Predictions through Solving IVPs Forward in Time

**Input**: Predicted Representation $\hat{\boldsymbol{r}}$

**Output**: One Patch Forecasts $\{\hat{\boldsymbol{x}}_i\}_{i=p_s}^{p_e}$

1: $\boldsymbol{z}_{p_s} = \hat{\boldsymbol{r}}$

2: $\{\Delta t_i\}_{i=p_s}^{p_e} = \{t_i\}_{i=p_s}^{p_e} - t_{p_s}$

3: $\{\boldsymbol{z}_i\}_{i=p_s}^{p_e} = \text{IVPSolve}(\boldsymbol{z}_{p_s}, \Delta t_i)\}_{i=p_s}^{p_e}$

4: $\{\hat{\boldsymbol{x}}_i\}_{i=p_s}^{p_e} = \text{Linear}(\{\boldsymbol{z}_i\}_{i=p_s}^{p_e})$

5: **return** $\{\hat{\boldsymbol{x}}_i\}_{i=p_s}^{p_e}$

---

This operation is simultaneously processed for all data points within the patch, yielding $\{\boldsymbol{z}_{0,i}\}_{i=1}^p$. From these, we infer a distribution $q(\boldsymbol{z}_0 \mid V)$, from which samples are drawn to produce the patch representation $\boldsymbol{r}$. In particular, we adopt the technique from variational autoencoders (VAEs) (Kingma & Welling, 2014) into the modeling of time series patches. Since the true posterior $p(\boldsymbol{z}_0 \mid V)$ is intractable to derive, we learn a variational approximation $q_\phi(\boldsymbol{z}_0 \mid V)$, from which $\boldsymbol{z}$ can be sampled. To compute $q_\phi(\boldsymbol{z}_0 \mid V)$ from

$\{q_\phi(\boldsymbol{z}_{0,i} \mid V)\}_{i=1}^p$ in Inference, we introduce a mixture of diagonal Gaussian distributions:

$$q_\phi(\boldsymbol{z}_{0,i} \mid V) = \mathcal{N}(\boldsymbol{\mu}_{\boldsymbol{z}_{0,i}}, \boldsymbol{\sigma}_{\boldsymbol{z}_{0,i}}), \tag{2}$$

where $\boldsymbol{\mu}_{\boldsymbol{z}_{0,i}}$ and $\boldsymbol{\sigma}_{\boldsymbol{z}_{0,i}}$ are parameterized by neural networks for each patch. In essence, for each input patch, we first derive a learned distribution, then sample from it to obtain the desired vector representations.

For the predictions in the subsequent patch (**Part 2**), the latent representation $\boldsymbol{hatz}_{p_s}$ produced by the LED Attentions at the end, is treated as the initial condition with timestamp $t_{p_s}$, to be evolved forward in time. This latent state is then advanced along the timeline to obtain state $\boldsymbol{z}_i$ using the same IVP solver, where $\Delta t_i = t_i - t_{p_s}$. A linear layer then maps $\boldsymbol{z}_i$ to the predicted observation values $\hat{\boldsymbol{x}}_i$.

### 3.2.3 LED ATTENTION

IVP Patcher primarily focuses on modeling the internal patterns within each patch. However, inter-patch correlations must also be effectively captured for accurate predictions. While the Transformer's self-attention mechanism is usually adopted for this purpose, this mechanism has been found to be less effective in capturing temporal order for time series data due to the limitations of their positional encoding functions (Zeng et al., 2023). To better accommodate time series with variable regularities, we adapt the Transformer self-attention layer into the LED Attention, incorporating the following novel design elements.

First, we enhance the temporal encoding technique in the LED Attention to model complex inter-patch dependencies with irregular time intervals. Inspired by rotary position embedding (RoPE) (Su et al., 2024) which offers strong theoretical properties and inductive bias for continuous values aligned with temporal information, we adopt this method and calculate Query and Key of the self-attention module by

$$f_{Q/K}(\boldsymbol{r}, \tau) = (\boldsymbol{W}\boldsymbol{r})\,e^{i\tau\theta}, \tag{3}$$

where $\boldsymbol{W}$ denotes a learnable transformation matrix, $i$ is the imaginary unit $i$, $\alpha$ is a non-zero constant, and $\tau$ denotes the patch time indicator. Since IVP Patcher is designed to model the evolution of patches with respect to their initial timestamps, the first timestamp within each patch is designated as the time indicator $\tau$. It is noteworthy that the patch time indicators $\{\tau_k\}_{k=0}^K$ may exhibit variable intervals, reflecting the irregular and continuous nature of the time series, making them well-suited to the RoPE technique.

In addition, we utilize the static domain information $L$, extracted from VT-Norm, by transforming this feature and appending it to the beginning of the input. This allows patches from any positions in the sequence to attend to domain-specific information during the autoregressive process. Furthermore, considering the LED Attention operates on the patch-level representations, we append a dummy patch at the end of the sequence, associated with the patch time indicator of the prediction horizon. This dummy patch experiences multiple interactions with preceding patches through the LED Attentions, achieving improved performance compared to late fusion of time information after LED Attentions when no dummy patch is used (4.2.4).

Finally, the input representations to the LED Attentions, denoted by $A_0$, can be constructed as $A_0 = [A_0^L, A_0^E, A_0^D]$, where $A_0^L$ is the transformed static domain feature from $L$, $A_0^E = \{\boldsymbol{r}_k\}_{k=1}^K$ denote the sequence of patch representations, with $K$ being the number of patches, and $A_0^D$ denotes the dummy patch representation. The concatenation is performed along the time dimension and processed through a stack of $m$ layers with LED Attention.

Afterwards, the patch representation is extracted at the dummy patch position: $\hat{\boldsymbol{r}} = A_m^T$, which is subsequently passed into neural IVP solvers to produce time series prediction results for the target horizon. Beyond these designs, the LED Attention maintains the original architecture of a decoder-only self-attention layer.

### 3.2.4 TRAINING

Since FlexTSF is essentially an autoregressive generative model, we train it by modeling the joint distribution of all data points in a sequence. Given the input sequence $\mathcal{X} = \{x_i\}_{i=1}^{M}$, the future sequence $\mathcal{X}^+ = \{x_i\}_{i=M+1}^{M+H}$, and the full sequence $\mathcal{X}^* = \{x_i\}_{i=1}^{M+H}$, the objective is to maximize $\log p(\mathcal{X}^*)$. By applying the Evidence Lower Bound (ELBO) (Kingma & Welling, 2014), it becomes:

$$\log p(\mathcal{X}^*) = \mathbb{E}_{\mathcal{Z}_0 \sim q_\phi(\mathcal{Z}_0|\mathcal{X}^*)}\left[\log p_\theta\left(\mathcal{X}^* \mid \mathcal{Z}_0\right)\right] - D_{KL}(q_\phi(\mathcal{Z}_0 \mid \mathcal{X}^*) \parallel p(\mathcal{Z}_0)) \tag{4}$$

$$\geq \mathbb{E}_{\mathcal{Z}_0 \sim q_\phi(\mathcal{Z}_0|\mathcal{X}^*)}\left[\log p_\theta\left(\mathcal{X}^+ \mid \mathcal{Z}_0\right)\right] - D_{KL}(q_\phi(\mathcal{Z}_0 \mid \mathcal{X}^*) \parallel p(\mathcal{Z}_0)) = -\mathcal{L} \tag{5}$$

where $\phi$ and $\theta$ are learnable parameters, and $\mathcal{Z}_0 = \{z_0^k\}_{k=1}^{K}$ represents the latent variables obtained from all patches. The first term corresponds to the log-likelihood of all available observations within the forecast horizon, while the second term is the KL-divergence between the learned posterior distribution of all patches and the prior distribution. Note that in $\mathcal{L}$, the likelihood is calculated based on the newly generated values $\mathcal{X}^+$ rather than $\mathcal{X}^*$, which is used in the calculation of the KL-divergence. The derivation and explanation of the loss can be found in Appendix A.2.

Using this framework, FlexTSF can be trained in both supervised and self-supervised settings (see Appendix A.4 for illustration). Moreover, after pre-training on a large collection of datasets, FlexTSF is found to be capable of achieving effective performance by fine-tuning on a target domain with only a small subset of parameters (parameters for transforming input, output values of time series, and static domain features). This attribute can significantly reduce the time and computational resources required to fine-tune the model.

## 4 EXPERIMENTS

### 4.1 EXPERIMENTAL SETTINGS

We evaluate FlexTSF's forecasting performance for both regular and irregular time series across different domains and training settings. We conduct three stages of experiments using two non-overlapping groups of datasets: the pre-training dataset $\mathcal{D}_p$ and the held-out dataset $\mathcal{D}_h$. In the first stage, we perform classic training-validation-testing experiments on $\mathcal{D}_h$ to demonstrate the effectiveness of FlexTSF. Next, we pre-train FlexTSF on $\mathcal{D}p$, yielding a pre-trained model with 61 million parameters. This model is initially used to perform zero-shot forecasts on the test partition of $\mathcal{D}_h$, evaluating its potential as a foundation model, and then fine-tuned on $\mathcal{D}_h$ for time series forecasting, assessing its adaptability to new domains in few-shot scenarios.

Each dataset in $\mathcal{D}_h$ is split into training, validation, and testing sets, following their original splits if known or a split ratio of 8:1:1, otherwise. As in prior works ((Ansari et al., 2024; Nie et al., 2023; Shukla & Marlin, 2021)), we repeat each experiment three times with different random seeds for dataset splitting and model parameter initialization. To ensure fair comparisons of model performance across datasets, we uniformly use the first 80% of each time series as input and the remaining 20% as the prediction target. The model performance is assessed using mean squared error (MSE). All models are tested in the same computing environment with NVIDIA Tesla V100 GPUs.

### 4.1.1 DATASETS

Our pre-training dataset group $\mathcal{D}_p$ consists of datasets from the Monash Time Series Forecasting Archive (Godahewa et al., 2021) and the UCR & UEA Time Series Classification Archive (Dau et al., 2019; Bagnall et al., 2018; Bagnall, 2024). After processing, $\mathcal{D}_p$ consists of 2.4 million sequences

with lengths ranging from 18 to 1024, spanning domains such as tourism, banking, energy, sales, economics, transportation, nature, web, and health. Our held-out dataset group $\mathcal{D}_h$ includes datasets from the Long Time Series Forecasting Benchmark (Lai et al., 2018; Wu et al., 2021) and the Irregular Time Series Benchmark (Rubanova et al., 2019; Li et al., 2018). We make two modifications to $\mathcal{D}_p$ and $\mathcal{D}_h$: first, we discard duplicate datasets. If a dataset appears in both $\mathcal{D}_p$ and $\mathcal{D}_h$, it is removed from $\mathcal{D}_h$.

| Category | Datasets |
|---|---|
| Regular time series | ETTh, ETTm, ExRate, Illness, Weather, HAR-Phone |
| Irregular Time series | |
| • *with missing values* | METR-LA, HAR-IMU |
| • *with various lengths* | ArabDigit, CharTraj |
| • *with unequal time intervals* | eICU, PhysioNet12 |

Table 2: Held-out datasets for comparing models

Second, to obtain a broadly representative held-out dataset $\mathcal{D}_h$, we move selected datasets from $\mathcal{D}_p$ to $\mathcal{D}_h$. Table 2 lists the datasets used in $\mathcal{D}_h$. Further details on the datasets can be found in Appendix A.5.

### 4.1.2 BASELINES

Previous studies have typically been conducted either on regular or irregular time series. Therefore, we incorporate baselines from both regular and irregular time series domains. To evaluate the zero-shot prediction capabilities of our pre-trained FlexTSF model, we include several pre-trained models. The baselines used in our research are listed in Table 3. One work (Darlow et al., 2024) considered irregularity when building a foundation model, but the model was not applied to irregular time series forecasting, and pre-trained checkpoints or source code to reproduce it have not been released. Therefore, we did not use it in the experiment. These three zero-shot baseline models originally do not support irregular data. To enable them to run on irregular datasets, we adapted the datasets by imputing missing values and aligning timestamps.

| Category | Baselines |
|---|---|
| Regular Time Series | RNN (Hochreiter & Schmidhuber, 1997), Transformer (Vaswani et al., 2017), Informer (Zhou et al., 2021), NHiTS (Challu et al., 2023), DLinear (Zeng et al., 2023), PatchTST (Nie et al., 2023) |
| Irregular Time Series | TaRNN (Rubanova et al., 2019), GRU-D (Che et al., 2018), MTAN (Shukla & Marlin, 2021), Latent-ODE (Shukla & Marlin, 2021), Latent-Flow (Biloš et al., 2021), ContiFormer (Chen et al., 2023) |
| Zero-shot Forecasts | Lag-Llama (Rasul et al., 2023), MOIRAI (Woo et al., 2024), TimesFM (Das et al., 2024) |

Table 3: Baselines used in the experiments.

### 4.2 RESULTS

#### 4.2.1 REGULAR & IRREGULAR TIME SERIES FORECASTING

Table 4 presents the mean squared error (MSE) results for our experiments in the classic training-validation-testing setting on $\mathcal{D}_h$. Across all datasets, FlexTSF consistently ranks among the best two models. On regular time series, FlexTSF performs comparably to state-of-the-art models, demonstrating its effectiveness in capturing temporal dynamics. For irregular datasets, FlexTSF excels, achieving the lowest MSE on METR-LA, CharTraj, and HAR-IMU, demonstrating its superiority when dealing with irregularity issues.

Among the baseline models, those designed for regular time series outperform other models on such time series, and their advanced designs enable some of them to also perform well on irregular datasets. The baselines specifically tailored to irregular time series perform particularly well on irregular time series, specifically on eICU and PhysioNet12, which have higher rates of missing data and greater sparsity.

#### 4.2.2 ZERO-SHOT FORECASTING

Table 5 shows results of our zero-shot experiments where we pre-train FlexTSF on $\mathcal{D}_p$ and then directly apply it to the testing data sets of $\mathcal{D}_h$. FlexTSF outperforms other pre-trained time series forecasting foundation models for all except one dataset, and performs exceptionally well on irregular datasets. Two of the baselines

| | | Regular Time Series | | | | | | Irregular Time Series | | | | | |
|---|---|---|---|---|---|---|---|---|---|---|---|---|---|
| | | ETTh | ETTm | ExRate | Illness | Weather | HAR-Phone | METR-LA | ArabDigit | CharTraj | HAR-IMU | eICU | PhysioNet12 |
| Regular | RNN | 0.913 | 0.845 | 0.834 | 4.417 | 0.213 | 0.521 | 0.424 | 0.672 | 0.406 | 0.551 | 0.931 | 0.848 |
| | Transformer | 1.169 | 1.010 | 0.436 | 4.867 | 0.133 | **0.419** | 0.391 | 0.502 | **0.135** | 0.282 | 0.559 | 0.496 |
| | Informer | 1.401 | 1.215 | 0.909 | 5.155 | 0.148 | 0.456 | 0.394 | 0.499 | 0.190 | 0.302 | 0.611 | 0.498 |
| | NHiTS | 2.703 | 2.600 | 2.333 | 4.397 | 0.597 | 0.850 | 1.047 | 0.799 | 0.194 | 0.804 | 0.914 | 0.839 |
| | DLinear | **0.218** | **0.169** | **0.035** | **3.058** | 0.150 | 0.558 | 0.410 | 0.718 | 0.554 | 0.513 | 0.689 | 0.556 |
| | PatchTST | 0.227 | 0.201 | 0.041 | 3.698 | **0.124** | 0.446 | **0.387** | 0.631 | 0.212 | **0.233** | 0.610 | 0.524 |
| Irregular | TaRNN | 0.894 | 0.615 | 0.775 | 4.118 | 0.148 | 0.481 | 0.411 | **0.460** | 0.177 | 0.496 | 0.601 | 0.518 |
| | GRU-D | 1.159 | 1.196 | 0.900 | 4.577 | 0.156 | 0.594 | 0.411 | 0.517 | 0.159 | 0.516 | 0.932 | 0.781 |
| | MTAN | 0.909 | 0.515 | 0.828 | 3.833 | 0.134 | 0.554 | 0.506 | 0.501 | 0.247 | 0.281 | **0.524** | **0.447** |
| | Latent-ODE | 2.419 | 1.658 | 1.012 | 4.206 | 0.175 | 0.610 | 0.446 | 0.759 | 0.951 | 0.395 | 0.528 | 0.614 |
| | Latent-Flow | 2.108 | 1.411 | 1.032 | 4.616 | 0.182 | 0.619 | 0.453 | 0.743 | 0.971 | 0.405 | 0.530 | 0.707 |
| | ContiFormer | 1.217 | 0.798 | 0.733 | 3.610 | 0.139 | 0.600 | 0.612 | 0.728 | 0.397 | 0.413 | 0.525 | 0.504 |
| | FlexTSF | **0.225** | **0.188** | **0.038** | **2.165** | **0.123** | **0.436** | **0.364** | **0.497** | **0.101** | **0.217** | **0.500** | **0.440** |

Table 4: Results of the classic training-validation-testing setting with FlexTSF and baselines specialised on regular and irregular time series measured with the Mean Squared Error (MSE). Top-2 results are **bolded**.

fail to produce meaningful results on eICU and PhysioNet12 within reasonable computing time and resources. FlexTSF's consistent performance across both regular and irregular datasets in zero-shot scenarios highlights its potential as a universal foundation model due to its native support for various data structures.

| | Regular Time Series | | | | | | Irregular Time Series | | | | | |
|---|---|---|---|---|---|---|---|---|---|---|---|---|
| | ETTh | ETTm | ExRate | Illness | Weather | HAR-Phone | METR-LA | ArabDigit | CharTraj | HAR-IMU | eICU | PhysioNet12 |
| Lag-Llama | 0.649 | 0.967 | 0.156 | 0.931 | 0.843 | 0.882 | 0.787 | 0.712 | 0.755 | 0.640 | NaN | NaN |
| MOIRAI | 0.267 | 0.470 | 0.282 | 1.805 | 0.246 | 0.715 | 0.659 | 0.841 | 0.879 | 0.526 | NaN | NaN |
| TimesFM | 0.344 | 0.741 | 0.127 | 2.729 | 0.539 | 0.948 | 0.741 | 0.803 | 1.150 | 0.857 | 0.996 | 1.143 |
| FlexTSF | 0.230 | 0.235 | 0.043 | 2.751 | 0.144 | 0.615 | 0.516 | 0.646 | 0.656 | 0.418 | 0.674 | 0.583 |

Table 5: Zero-shot forecasting performance comparison between FlexTSF and other pre-trained models on regular and irregular time series datasets. The table reports Mean Squared Error (MSE) values. 'NaN' values indicate that the model is not applicable to that dataset.

### 4.2.3 FEW-SHOT FORECASTING

Figure 3 shows results of our few-shot forecasting experiments: after pre-training on $\mathcal{D}_p$, we fine-tune FlexTSF using varying numbers of samples drawn from the training sets of $\mathcal{D}_h$ and evaluate it on the corresponding test sets. FlexTSF consistently outperforms all baselines across both regular and irregular datasets, with even greater advantages when fewer training samples are available. This underscores FlexTSF's superior sample efficiency and adaptability, making it highly effective in few-shot learning scenarios where data is limited.

### 4.2.4 ABLATION STUDY

To assess the contributions of our components VT-Norm, IVP Patcher, and LED Attention, we conduct an ablation study by independently removing each component at a time, resulting in three different model variants. Each model is pre-trained on the same datasets ($D_p$) using the same computational resources. After pre-training, all models are evaluated through zero-shot forecasting on the same test datasets. Table 6 presents the MSE changes compared to the original FlexTSF model in zero-shot scenarios.

Removing VT-Norm results in clear performance degradation, particularly on ExRate and HAR-IMU, highlighting its importance in handling domain-specific variations by decoupling static and dynamic patterns. The absence of the IVP Patcher causes severe degradation on irregular datasets such as CharTraj and ArabDigit,

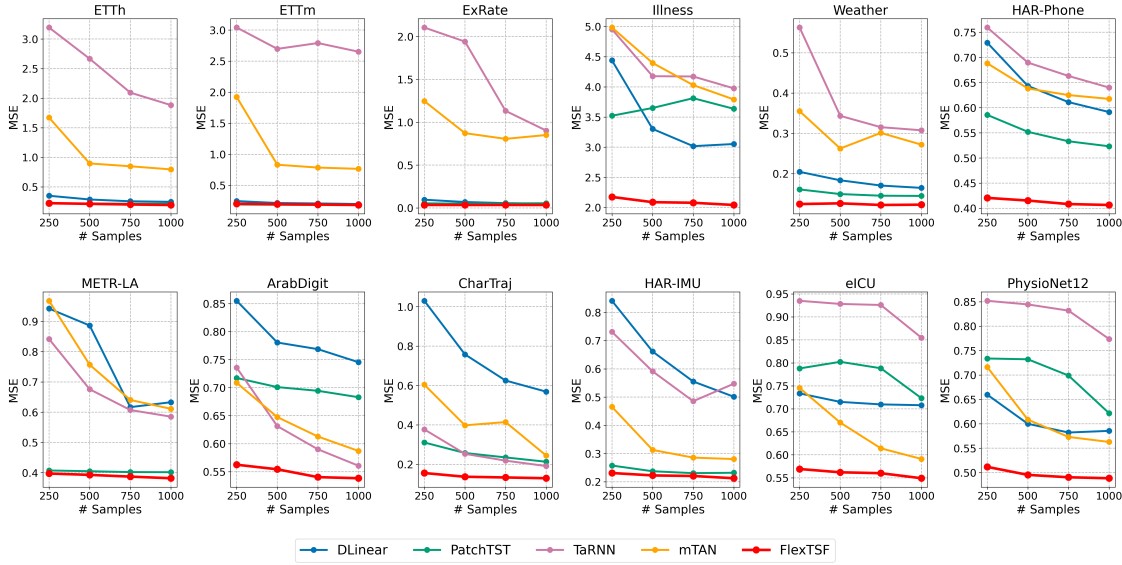

Figure 3: Results of our few-shot experiments. The x-axis shows the number of samples used for fine-tuning, and the y-axis presents the Mean Squared Error (MSE) across different datasets.

demonstrating its essential role in handling irregular intervals and missing values. Removing LED Attention also leads to increased MSE for all datasets, particularly on CharTraj and Weather. This suggests that LED Attention is also indispensable due to its role in encapsulating static domain information, time information, and value series within the autoregressive forecasting architecture.

| Left-out | | Regular Time Series | | | | | | Irregular Time Series | | | | | |
|---|---|---|---|---|---|---|---|---|---|---|---|---|---|
| | | ETTh | ETTm | ExRate | Illness | Weather | HAR-Phone | METR-LA | ArabDigit | CharTraj | HAR-IMU | eICU | PhysioNet12 |
| VT-Norm | MSE | 0.969 | 0.983 | 1.322 | 4.436 | 0.24 | 0.593 | 0.629 | 0.64 | 0.788 | 0.731 | 1.211 | 1.157 |
| | MSE+/- | +321.30% | +318.30% | +2974.42% | +61.25% | +66.67% | -3.58% | +21.90% | -0.93% | +20.12% | +74.88% | +79.67% | +98.46% |
| IVP Patcher | MSE | 0.238 | 0.27 | 0.039 | 13.824 | 0.146 | 0.711 | 0.407 | 1.606 | 2.089 | 1.675 | 1.063 | 0.884 |
| | MSE+/- | +5.78% | +43.62% | +2.63% | +538.52% | +18.70% | +63.07% | +11.81% | +223.14% | +1968.32% | +671.89% | +112.60% | +100.91% |
| LED Attention | MSE | 0.263 | 0.21 | 0.039 | 3.107 | 0.157 | 0.696 | 0.443 | 0.654 | 0.716 | 0.291 | 0.895 | 0.738 |
| | MSE+/- | +16.89% | +11.70% | +2.63% | +43.51% | +27.64% | +59.63% | +21.70% | +31.59% | +608.91% | +34.10% | +79.00% | +67.73% |

Table 6: Ablation study results for FlexTSF. The MSE row presents the mean squared error across all datasets. The MSE+/- row shows the change in MSE compared to the zero-shot results of FlexTSF in Table 5.

## 5 CONCLUSION

We introduced FlexTSF, a universal time series forecasting model designed to address the challenges posed by the domain and structural diversity of time series. FlexTSF comes with three novel designs, namely VT-Norm, IVP Patcher, and LED Attention, which together enable the model to generalize across diverse domains and handle irregularities in time series data. Our experiments across 12 datasets demonstrate that FlexTSF consistently outperforms state-of-the-art models specifically designed for either regular or irregular time series. FlexTSF's ability to excel in both few-shot and zero-shot settings highlights its versatility as a foundation model. In future work, the impact of pre-training on datasets with larger scale could be further explored to push the boundaries of foundation models in time series forecasting.

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

# A APPENDIX

## A.1 NOTATIONS

An overview of notations used in this paper is given in Table 7.

## A.2 DERIVATION OF THE LOSS FUNCTION

Given $\mathcal{X} = \{\boldsymbol{x}_i\}_{i=1}^{M}$, $\mathcal{X}^+ = \{\boldsymbol{x}_i\}_{i=M+1}^{M+H}$ and $\mathcal{X}^* = \{\boldsymbol{x}_i\}_{i=1}^{M+H}$, following ELBO in VAE (Kingma & Welling, 2014), we can obtain the standard formation as:

$$\log p(\mathcal{X}^*) = \mathbb{E}_{\mathcal{Z}_0 \sim q_\phi(\mathcal{Z}_0 | \mathcal{X}^*)} \left[\log p_\theta \left(\mathcal{X}^* \mid \mathcal{Z}_0\right)\right] - D_{KL}(q_\phi(\mathcal{Z}_0 \mid \mathcal{X}^*) \parallel p(\mathcal{Z}_0))$$

where for the likelihood, we have:

$$\log p_\theta \left(\mathcal{X}^* \mid \mathcal{Z}_0\right) = \log p_\theta \left(\mathcal{X} \cdot \mathcal{X}^+ \mid \mathcal{Z}_0\right) = \log p_\theta \left(\mathcal{X} \mid \mathcal{X}^+, \mathcal{Z}_0\right) + \log p_\theta \left(\mathcal{X}^+ \mid \mathcal{Z}_0\right)$$

We further have:

$$\log p_\theta \left(\mathcal{X}^* \mid \mathcal{Z}_0\right) \geq \log p_\theta \left(\mathcal{X}^+ \mid \mathcal{Z}_0\right)$$

By substituting this term into ELBO, we obtain:

$$\log p(\mathcal{X}^*) \geq \mathbb{E}_{\mathcal{Z}_0 \sim q_\phi(\mathcal{Z}_0 | \mathcal{X}^*)} \left[\log p_\theta \left(\mathcal{X}^+ \mid \mathcal{Z}_0\right)\right] - D_{KL}(q_\phi(\mathcal{Z}_0 \mid \mathcal{X}^*) \parallel p(\mathcal{Z}_0))$$

Then the question follows, can we maximize $\log p_\theta \left(\mathcal{X}^* \mid \mathcal{Z}_0\right)$ by maximizing $\log p_\theta \left(\mathcal{X}^+ \mid \mathcal{Z}_0\right)$? The answer is Yes, with some presumption. The fundamental assumption of all time series forecasting models is that future values $\mathcal{X}^+$ can be predicted based on past values $\mathcal{X}$. To achieve this, researchers often further assume that $\mathcal{X}$ and $\mathcal{X}^+$ follow the same distribution. This distribution can be explicit, such as a Gaussian distribution, which leads to the concept of stationarity, where statistical properties—such as mean, variance, and autocovariance—remain constant over time. Alternatively, the distribution can be implicit or unknown, in which case it can be learned by neural networks. In this work, we adopt the same assumption. From this perspective, if a neural network is good at modeling $\mathcal{X}^+$, then it is good at modeling $\mathcal{X}^*$. Modeling either can be used to train a neural network. We chose to model $\mathcal{X}^+$ because it is more efficient.

| Notation | Description |
|---|---|
| $S = \{(\boldsymbol{x}_i, t_i)\}_{i=1}^{M}$ | A time series |
| $M$ | Number of observations in a time series |
| $\boldsymbol{x}_i$ | The observation values |
| $\mathcal{X}$ | Observed value sequence $\boldsymbol{x}_{t_1:t_M}$ |
| $\mathcal{X}^+$ | Forecasted value sequence $\boldsymbol{x}_{t_{M+1}:t_{M+H}}$ |
| $T$ | All the timestamps of $\mathcal{X}$ |
| $t_i$ | Corresponding timestamp for each $\boldsymbol{x}_i$ |
| $v$ | The number of variables or channels |
| $H$ | The forecast horizon |
| $D$ | A dataset |
| $\mu_g$ | The global mean of a dataset |
| $\sigma_g$ | The global standard deviation of a dataset |
| $G_g$ | The global normalization operation |
| $\mu_i$ | The instance mean of a time series |
| $\sigma_i$ | The instance standard deviation of a time series |
| $G_i$ | The instance normalization operation |
| $\omega_g$ | The global time unit |
| $\omega_i$ | The instance time unit |
| $t_i'/T'$ | The time quantity/quantities obtained through normalization |
| $L$ | The extracted static information |
| $V = \{(\boldsymbol{x}_i, t_i)\}_{i=1}^{p}$ | A time series patch |
| $\boldsymbol{z}_i$ | The hidden state of $\boldsymbol{x}_i$ |
| $p_s$ | The starting index of a patch |
| $p_e$ | The ending index of a patch |
| $\boldsymbol{r}$ | Representation of a patch |
| $\hat{\boldsymbol{r}}$ | Predicted representation of a patch |
| $\hat{\boldsymbol{x}}_i$ | Predicted observation values |
| $\tau$ | The time indicator for a patch |
| $A_0$ | The input of the first attention layer |
| $A_0^L$ | Part of the input, produced by the static information $L$ |
| $A_0^E$ | Part of the input, produced by time series patches |
| $A_0^D$ | Part of the input, produced by the dummy patch |
| $K$ | The number of patches |
| $m$ | Number of attention layers |
| $\mathcal{L}$ | Loss function |
| $\mathcal{Z}_0$ | Latent variables obtained from all patches |
| $\phi$ | Recognition model of VAE i.e., Input IVP Patcher in FlexTSF |
| $\theta$ | Generation model of VAE i.e., LED Attention and Output IVP Patcher |
| $\Gamma$ | All the learnable parameters |
| $\mathcal{D}_h$ | The held-out dataset group |
| $\mathcal{D}_p$ | The pre-trained dataset group |

Table 7: Notation table.

If we are about to model $\mathcal{X}^*$, based on the chain rule, we have:

$$\mathbb{P}(\boldsymbol{x}_1 \cap \boldsymbol{x}_2 \cap \cdots \cap \boldsymbol{x}_{M+N}) = \mathbb{P}(\boldsymbol{x}_{M+N} \mid \boldsymbol{x}_1 \cap \cdots \cap \boldsymbol{x}_{M+N-1}) \ldots \mathbb{P}(\boldsymbol{x}_3 \mid \boldsymbol{x}_1 \cap \boldsymbol{x}_2) \mathbb{P}(\boldsymbol{x}_2 \mid \boldsymbol{x}_1) \mathbb{P}(\boldsymbol{x}_1)$$

$$= \prod_{i=1}^{M+N} \mathbb{P}(\boldsymbol{x}_i \mid \boldsymbol{x}_1 \cap \cdots \cap \boldsymbol{x}_{i-1}).$$

However, if we about to model $\mathcal{X}^+$, based on the chain rule, we have:

$$\mathbb{P}(\boldsymbol{x}_{M+1} \cap \cdots \cap \boldsymbol{x}_{M+N} \mid \boldsymbol{x}_{1:M}) = \mathbb{P}(\boldsymbol{x}_{M+N} \mid \boldsymbol{x}_1 \cap \cdots \cap \boldsymbol{x}_{M+N-1}) \ldots \mathbb{P}(\boldsymbol{x}_{M+1} \mid \boldsymbol{x}_1 \cap \cdots \cap \boldsymbol{x}_{M+N-1})$$

$$= \prod_{i=M+1}^{M+N} \mathbb{P}(\boldsymbol{x}_i \mid \boldsymbol{x}_1 \cap \cdots \cap \boldsymbol{x}_{i-1}).$$

The latter is more efficient. Since attention modules are often computationally intensive, this efficiency advantage benefits the model during both training and inference.

### A.3  MORE RELATED WORKS

#### A.3.1  TRANSFORMERS FOR TIME SERIES FORECASTING

Time series forecasting has been a longstanding research area, evolving from traditional statistical approaches such as ARIMA (Box & Jenkins, 1970) and GARCH (Bollerslev, 1986) to modern deep learning models based on RNNs (Salinas et al., 2020), CNNs (Borovykh et al., 2018), and Transformers (Zhou et al., 2021). In recent years, the powerful sequential modeling capabilities of Transformers have led to the development of numerous Transformer-based models (Li et al., 2019; Zhou et al., 2021; Wu et al., 2021; Zhou et al., 2022; Liu et al., 2021), with an emphasis on reducing the computational complexity of the original attention mechanism to enhance the feasibility of long-term time series forecasting. Notably, PatchTST (Nie et al., 2023) represents a significant advancement by employing patch-level representations instead of processing individual records at each timestamp. This strategy has become a cornerstone, which has proven effective in capturing complex temporal patterns and improving forecasting accuracy.

#### A.3.2  FOUNDATION MODELS FOR TIME SERIES FORECASTING

Building on the remarkable successes of large language models (LLMs) across various domains, several studies have attempted to adapt LLMs to the domain of time series forecasting. PromptCast (Xue & Salim, 2023) transforms numerical input into prompts, and frames the forecasting task in a conversational manner. GPT4TS (Zhou et al., 2023) and Chronos (Ansari et al., 2024) finetune GPT-2 (Radford et al., 2019) and T5 (Raffel et al., 2020) by directly training them on time series data for respective dataset or applying time series tokenization. Time-LLM (Jin et al., 2024) is a reprogramming method that aligns input time series data with text prototypes, and transforms them into prefix prompts to augment LLM's ability to reason with time series data. TEMPO (Cao et al., 2024) decomposes time series into trend, seasonal, and residual components and designs prompts for distribution adaptation. TEST (Sun et al., 2024) employs contrastive learning to align time series representations with the language space of LLM. More related to our model is ISTS-PLM (Zhang et al., 2024), which adapts LLM to handle irregularly sampled time series.

#### A.3.3  IRREGULAR TIME SERIES MODELING

In addition to regular time series, which have uniformly sampled data points, irregular time series are becoming increasingly prevalent due to widespread adoption of various sensing devices and recording practices (Weerakody et al., 2021). In recent years, substantial progress has been made in developing models to handle irregular time series, as demonstrated by works such as GRU-D (Che et al., 2018), Latent-ODE

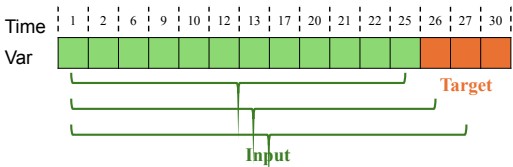
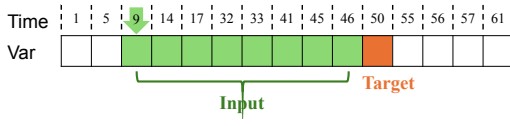

(a) Input and target during training & inference.
(b) Input and target during pre-training.

Figure 4: Time series from different domains.

(Rubanova et al., 2019), mTAN (Shukla & Marlin, 2021), and IVP-VAE (Xiao et al., 2024). However, research on regular and irregular time series has largely progressed in parallel. A recent attempt (Chen et al., 2023) to bridge this gap proposes a model capable of handling both regular and irregular time series, but it suffers from high memory consumption and inefficient performance. Moreover, there remains a scarcity of foundational models that can be seamlessly applied across both data types. Irregularity is considered in (Darlow et al., 2024) when building a foundation model, but the model was not applied to irregular time series forecasting.

## A.4 ILLUSTRATION OF THE TRAINING PROCESS

As illustrated in Figure 4, to simplify the description, let's assume we have one variable in the time series, the patch length is 1, and observations are collected at irregular timestamps. In the supervised training or inference phase, each forward pass generates the next patch, which is then used as input to predict subsequent values until the entire forecast horizon is populated. During unsupervised pre-training, instead of using the full sequence, a sub-sequence is randomly selected at each iteration. First, a starting position is randomly determined, followed by the random selection of the sub-sequence length. The goal of pre-training is to predict the observations that follow the sub-sequence.

## A.5 DATASET DETAILS

Table 8 gives an overview of all datasets using in $\mathcal{D}_h$ and their characteristics. Here, the min time unit is measured in seconds, which describes the smallest sampling interval of the time series data, i.e., the $\omega_g$ in the model. The sampling interval for regular time series data is consistent, while for irregular time series, the sampling interval of some data may be larger than the minimum interval. The sequence length refers to the real length used in our experiments. For irregular time series data, this length represents its maximum length.

## A.6 HYPERPARAMETER AND MODEL SETTINGS

**Classic Training**

- Optimizer: Adam

---

[2] ETTh, ETTm, ExRate, Illness, Weather: https://github.com/thuml/Autoformer
[3] https://archive.ics.uci.edu/dataset/240/human+activity+recognition+using+smartphones
[4] https://github.com/liyaguang/DCRNN
[5] https://archive.ics.uci.edu/dataset/195/spoken+arabic+digit
[6] https://archive.ics.uci.edu/dataset/175/character+trajectories
[7] https://archive.ics.uci.edu/dataset/196/localization+data+for+person+activity
[8] https://physionet.org/content/eicu-crd/2.0/
[9] https://physionet.org/content/challenge-2012/1.0.0/

| Dataset (Full Name) | Domain | # Variables | Min Time Unit (s) | Sequence Length |
|---|---|---|---|---|
| ETTh (ETTh2)[2] | Power Systems | 7 | 3600 | 192 |
| ETTm (ETTm2) | Power Systems | 7 | 900 | 512 |
| ExRate (Exchange Rate) | Finance | 8 | 86400 | 192 |
| Illness | Epidemiology | 7 | 604800 | 96 |
| Weather | Meteorology | 21 | 600 | 256 |
| HAR-Phone[3] | Wearable Computing | 9 | 0.02 | 128 |
| METR-LA[4] | Traffic | 207 | 86400 | 24 |
| ArabDigit (SpokenArabicDigits)[5] | Speech Recognition | 13 | 0.000091 | 93 |
| CharTraj (CharacterTrajectories)[6] | Handwriting Recognition | 3 | 0.005 | 182 |
| HAR-IMU[7] | Human Activity Recognition | 12 | 0.1 | 50 |
| eICU[8] | Healthcare | 14 | 60 | 446 |
| PhysioNet12[9] | Healthcare | 37 | 60 | 216 |

Table 8: Characteristics of the datasets in $\mathcal{D}_h$.

- LR Scheduler: StepLR
- Weight decay: 1e-4
- Batch size: 64
- Learning rate: 1e-4
- Learning rate scheduler step: 20
- Learning rate decay: 0.5

**Pre-Training**

- Optimizer: AdamW
- LR Scheduler: CosineAnnealingLR
- Warmup-steps: 1000
- Batch size: 64
- Weight Decay: 0.1
- beta1: 0.9
- beta2: 0.95

**Fine-tuning**

- Optimizer: Adam
- Weight decay: 1e-4
- Batch size: 64
- Learning rate: 1e-4
- Learning rate scheduler step: 20
- Learning rate decay: 0.5

**FlexTSF for Classic Training**

- Dimension of Each Head: 16
- Number of Heads: 4
- Number of Attention Layers: 2
- Dimension of IVP Solver: 64
- Total parameters: 440,066

**FlexTSF for Pre-Training, Zero-Shot, Fine-Tuning**

- Dimension of Each Head: 64
- Number of Heads: 12
- Number of Attention Layers: 6
- Dimension of IVP Solver: 768
- Total parameters: 61,488,514

