# OpenReview forum: "FlexTSF: A universal forecasting model for time series with variable regularities"
_ICLR.cc/2025/Conference — Submitted to ICLR 2025_

### Official Review · Reviewer_3hea · 2024-10-29

**Soundness:** 3
**Presentation:** 2
**Contribution:** 2
**Rating:** 3
**Confidence:** 4

**Summary:**

The paper proposes a  universal time series forecasting model FlexTSF that possesses better generalization and natively support both regular and irregular time series. Experiments show that FlexTSF demonstrates exceptional performance in both zero-shot and few-show settings for time series forecasting and outperforms stateof-the-art forecasting models respectively designed for regular and irregular time series.

**Strengths:**

1. The idea of a unified forecasting model for time series with variable regularities is interesting.
2. Experimental results demonstrate FlexTSF’s ability to excel in both few-shot and zero-shot settings highlighting its versatility as a
foundation model.

**Weaknesses:**

1. Some notations need further clarification. For example, the notation $(x_i, t_i)$ in Line 133, do the two subscripts $i$ share the same meaning? The notation $\Delta t$ seems been multi-defined in Line 187, Line 211, and Line 229.
2. Some highly relevant related works are missing, such as iTransformer for regular time series data and CRU for irregular time series data.
[1] Liu, Yong, et al. "itransformer: Inverted transformers are effective for time series forecasting." ICLR 2023.
[2] Schirmer, Mona, et al. "Modeling irregular time series with continuous recurrent units." ICML 2022.
3. The paper does not provide a detailed experimental setup, especially the forecast horizon. It also lacks a specific explanation for why the sequence length is set to 192 for ETTh2 and 512 for ETTm2 and why the held-out dataset does not include ETTh1, ETTm1, ECL, and Traffic datasets.

**Questions:**

Please refer to Weaknesses.

---

### Official Review · Reviewer_nbMZ · 2024-11-03

**Soundness:** 2
**Presentation:** 2
**Contribution:** 3
**Rating:** 5
**Confidence:** 5

**Summary:**

The article tackles two main issues in time series data: domain diversity and structural diversity, proposing the FlexTSF model. FlexTSF uses VT-norm to help the model distinguish between different domain data and employs IVT-patcher to handle irregular time series. It also introduces LED attention to capture relationships between time series, enhancing the model's capabilities. Experimental results show that FlexTSF performs well on both regular and irregular time series and demonstrates effective few-shot and zero-shot learning.

**Strengths:**

- The article innovatively uses IVT-patcher as a tokenization approach for time series, enabling foundational models to handle irregular time series data.
- The article conducts ablation experiments on each component, demonstrating that all components contribute positively to the model's performance.

**Weaknesses:**

**Experimental Setup Issues**

- The article does not provide a detailed description of the experimental setup, including input and output lengths, which significantly affects the results. This lack of information makes it challenging for reviewers to compare the results intuitively with previous works in temporal prediction, ultimately diminishing the credibility of the model's effectiveness.
- Additionally, the paper's baselines do not include recent significant contributions in temporal foundational models, such as Timer[1], Moment[2], TTM[3], and Time-MOE[4].

**The novelty of the article is somewhat limited.**

- **VT-norm:**
    - The two-stage normalization essentially combines overall normalization and instance normalization, which are common techniques in time series processing.
    - The authors concatenate all features into a vector to feed into the model as a prompt, claiming this approach addresses domain diversity. However, this method has been utilized in various studies, such as TimeLLM[5], which encodes statistical and domain information into prompts for LLMs. Moreover, relying solely on statistical information from the dataset does not effectively resolve the issue of domain diversity, as different domains may share similar statistical properties.
- **LED Attention:**
    - **Leader Node:** The Leader Node can be viewed as a prompt based on statistical information from the dataset, which has also been used in many other models.
    - **Dummy Patch:** This concept is similar to the decoder inputs in previous models like Informer[6] and Autoformer[7].
    - **Positional Encoding:** The approach is simply a straightforward application of RoPE.

**Reference**

[1] Liu, Y., Zhang, H., Li, C., Huang, X., Wang, J., & Long, M. (2024). Timer: Transformers for time series analysis at scale. arXiv preprint arXiv:2402.02368.

[2] Goswami, M., Szafer, K., Choudhry, A., Cai, Y., Li, S., & Dubrawski, A. (2024). Moment: A family of open time-series foundation models. arXiv preprint arXiv:2402.03885.

[3] Ekambaram, V., Jati, A., Nguyen, N. H., Dayama, P., Reddy, C., Gifford, W. M., & Kalagnanam, J. (2024). TTMs: Fast Multi-level Tiny Time Mixers for Improved Zero-shot and Few-shot Forecasting of Multivariate Time Series. arXiv preprint arXiv:2401.03955.

[4] Shi, X., Wang, S., Nie, Y., Li, D., Ye, Z., Wen, Q., & Jin, M. (2024). Time-MoE: Billion-Scale Time Series Foundation Models with Mixture of Experts. arXiv preprint arXiv:2409.16040.

[5] Jin, M., Wang, S., Ma, L., Chu, Z., Zhang, J. Y., Shi, X., ... & Wen, Q. (2023). Time-llm: Time series forecasting by reprogramming large language models. *arXiv preprint arXiv:2310.01728*.

[6] Wu, H., Xu, J., Wang, J., & Long, M. (2021). Autoformer: Decomposition transformers with auto-correlation for long-term series forecasting. *Advances in neural information processing systems*, *34*, 22419-22430.

[7] Zhou, H., Zhang, S., Peng, J., Zhang, S., Li, J., Xiong, H., & Zhang, W. (2021, May). Informer: Beyond efficient transformer for long sequence time-series forecasting. In *Proceedings of the AAAI conference on artificial intelligence* (Vol. 35, No. 12, pp. 11106-11115).

**Questions:**

- Could the authors provide detailed experimental setups, including the input and prediction lengths for each model across the datasets, as well as a comparison table with previous mainstream temporal prediction works? Specifically, the prediction lengths should be (96, 192, 336, 720) to facilitate intuitive comparisons for reviewers.
- The authors need to include recent works on time series foundational models to better substantiate the effectiveness of your model.

---

### Official Review · Reviewer_mWhM · 2024-11-04

**Soundness:** 3
**Presentation:** 2
**Contribution:** 2
**Rating:** 5
**Confidence:** 4

**Summary:**

This paper aims at developing a "flexible" time-series transformer capable of producing forecasts for both regularly and irregularly sampled time series.The developed FlexTSF model encompasses a specific normalization (VT-Norm), a customized patching module (IVP Patcher), LED Attention, and VAE-style self-supervised training.

**Strengths:**

- Unified forecasting for regular and irregular time-series is a good contribution.

**Weaknesses:**

I have several critical concerns, which are summarized below.

1. Seemingly Complicated Designs, Yet Essentially Combinations of Existing Techniques

This paper introduces several new technical names that are essentially derived from existing studies. For instance, VT-Norm employs a global preprocessing method for time-series values, followed by per-batch instance normalization, and also normalizes timestamps. IVP Patcher utilizes a technique from another study that solves initial value problems (IVP) to compute the patch embedding. LED Attention is essentially causal self-attention, but with some additional "tokens" as inputs, raising the question of why a new name for attention is necessary. The VAE-style training is akin to existing studies like mTAN. I suggest that the authors include techniques from existing studies in a preliminary section and focus on their unique designs and adaptations in the methodology section.

2. The Rationale of Using Irregular Time Series to Boost Regular Forecasting

I understand that traditional models for regular time series do not perform well on irregular time series due to sampling irregularities. However, it's unclear how irregular time series and the derived architectures improve performance on regular time series. For instance, Table 5 is confusing to me because pre-trained time-series models significantly underperformed compared to FlexTSF on regular time-series benchmarks. Is this due to differences in pre-training data or the unique design of FlexTSF? Unfortunately, I don't find a specific explanation for these performance differences, leading me to question the robustness of the experimental process.

3. Lack of Verifications on Irregular Time-series Classification and Comparisons with Other Available Baselines

In the context of pre-training irregular time series, it's crucial to verify the representation learning capability of FlexTSF on typical irregular time-series classification tasks. Most irregular time-series models in the literature are evaluated based on their ability to learn effective representations for classification tasks, particularly in healthcare scenarios. For example, irregular time-series baselines such as GRU-D, mTAND, Latent-ODE, and ContiFormer follow this approach. In this thread, more recent studies, such as Warpformer [1], are not included.

Addtionally, a published study focusing on irregular time-series forecasting [2] provides a comprehensive comparison of various models applicable to irregular time-series forecasting.

[1] Warpformer: A Multi-scale Modeling Approach for Irregular Clinical Time Series, https://dl.acm.org/doi/10.1145/3580305.3599543

[2] Irregular Multivariate Time Series Forecasting:A Transformable Patching Graph Neural Networks Approach, https://openreview.net/pdf?id=UZlMXUGI6e

**Questions:**

- Please provide a detailed rationale for how leveraging irregular time series can improve forecasting for regular time series. It is valuable to provide a detailed analysis of the factors contributing to FlexTSF's superior performance on regular time series in Table 5, including potential impacts of pre-training data, model architecture, and other relevant factors.
- When the focus is on irregular time series, to what extent does incorporating regular time series data contribute to any observed performance improvements? Providing specific data or case studies that highlight this contribution would be valuable.
- Including more recent baselines and representative tasks on irregular time series. For example, you may consider to include some representative baselines from [1], such as T-PatchGNN, Warpformer, RainDrop, SeFT, and also include addtional datasets, such as MIMIC.

[1] Irregular Multivariate Time Series Forecasting:A Transformable Patching Graph Neural Networks Approach, https://openreview.net/pdf?id=UZlMXUGI6e

---

### Official Review · Reviewer_5Z12 · 2024-11-06

**Soundness:** 3
**Presentation:** 3
**Contribution:** 2
**Rating:** 6
**Confidence:** 4

**Summary:**

The paper introduces FlexTSF, a time series forecasting model designed to handle regular and irregular time series. Unlike models limited to regularly structured data, FlexTSF supports variable sequence lengths, missing values, and irregular time intervals. The model has three main components: (1) VT-Norm, a normalization technique to manage domain diversity, (2) IVP Patcher, a patching module to handle structural diversity, and (3) LED Attention, a causal attention mechanism that integrates domain and temporal information for autoregressive forecasting. Results show that FlexTSF performs well on 12 datasets, in zero-shot and few-shot scenarios.

**Strengths:**

* FlexTSF addresses an important problem by extending time series forecasting to irregularly structured data. Its integration of VT-Norm, IVP Patcher, and LED Attention showcases is well-motivated.

* The model is well-implemented, with comprehensive testing on 12 diverse datasets covering regular and irregular time series. The authors validate their approach through extensive quantitative evaluations, including zero-shot and few-shot learning scenarios.

* The writing is clear, and each of the three key model components is clearly motivated. Each component of FlexTSF is well-explained, and the modular approach (VT-Norm, IVP Patcher, LED Attention) provides a clear framework for readers to understand the model’s mechanics.

**Weaknesses:**

*  Although FlexTSF is evaluated against state-of-the-art baselines, the comparison could be extended to include recent foundational models in time series, such as UniTS (NeurIPS 2024) and Reprogrammed LLMs for time series. Inclusion of these models would provide a more comprehensive benchmark. There are forecasting models missing from the comparison, such as DeepAR for probabilistic forecasting and Autoformer for long-sequence time series forecasting.

* The paper centers exclusively on forecasting as the primary task. Expanding FlexTSF to support other time series tasks—such as classification, anomaly detection, and imputation—could broaden its usability. These tasks are highly relevant in domains where time series models often need to operate on diverse tasks, and demonstrating FlexTSF across time series tasks would make the model more compelling as a foundation model.

* A combination of IVP Patcher and LED Attention make interpreting model outputs challenges. Additionally, using neural IVP solvers and LED Attention layers can be computationally intensive. It would be helpful to profile the computational load FlexTSF.

* FlexTSF’s approach does not explicitly address and is tested under forecasting with variable horizon lengths, where the desired prediction window may change dynamically based on the application. This could limit the model’s applicability in settings where adaptive forecasting horizons are needed, such as in finance and emergency response.

**Questions:**

Please see the list of weaknesses above.

---

### Meta-Review · Area_Chair_CAtn · 2024-12-19

**Metareview:**

This paper proposes a neural network architecture for time-series. It comprises a global and instance-specific normalization, these representations are then fed into IVPPatcher, a layer that leverages the identification of solutions to Initial Value Problems -- the intuition behind this is that the identification of the initial values under dynamical systems can be a useful sufficient statistic for representation learning for time-series data, this is then incorporated into an attention layer (LED attention). I think scaling the study of IVPsolvers to learn representations of time-series data is a very interesting direction. However, the initial reviews identified several open questions such as comparison to relevant baselines, reformatting the paper so that prior work was relegated to the background section and the relative improvements were the focus of the methodology as well as performing ablation studies to understand the impact of regular time series on learning representations of irregular ones.

**Additional Comments On Reviewer Discussion:**

The authors did not provide a response to the original set of reviews.

---

### Decision · Program_Chairs · 2025-01-22

Reject